# Identification of Fungicide Combinations Targeting *Plasmopara viticola* and *Botrytis cinerea* Fungicide Resistance Using Machine Learning

**DOI:** 10.3390/microorganisms11051341

**Published:** 2023-05-19

**Authors:** Junrui Zhang, Sandun D. Fernando

**Affiliations:** Biological and Agricultural Engineering Department, Texas A&M University, College Station, TX 77843-2117, USA

**Keywords:** cytochrome b, QoI fungicides, fungicide resistance, QSAR, machine learning, grapes, downy mildew, gray mold

## Abstract

Downy mildew (caused by *Plasmopara viticola*) and gray mold (caused by *Botrytis cinerea*) are fungal diseases that significantly impact grape production globally. Cytochrome b plays a significant role in the mitochondrial respiratory chain of the two fungi that cause these diseases and is a key target for quinone outside inhibitor (QoI)-based fungicide development. Since the mode of action (MOA) of QoI fungicides is restricted to a single active site, the risk of developing resistance to these fungicides is deemed high. Consequently, using a combination of fungicides is considered an effective way to reduce the development of QoI resistance. Currently, there is little information available to help in the selection of appropriate fungicides. This study used a combination of in silico simulations and quantitative structure–activity relationship (QSAR) machine learning algorithms to screen the most potent QoI-based fungicide combinations for wild-type (WT) and the G143A mutation of fungal cytochrome b. Based on in silico studies, mandestrobin emerged as the top binder for both WT *Plasmopara viticola* and WT *Botrytis cinerea* cytochrome b. Famoxadone appeared to be a versatile binder for G143A-mutated cytochrome b of both *Plasmopara viticola* and *Botrytis cinerea.* Thiram emerged as a reasonable, low-risk non-QoI fungicide that works on WT and G143A-mutated versions of both fungi. QSAR analysis revealed fenpropidin, fenoxanil, and ethaboxam non-QoIs to have a high affinity for G143A-mutated cytochrome b of *Plasmopara viticola* and *Botrytis cinerea*. Above-QoI and non-QoI fungicides can be considered for field studies in a fungicide management program against *Plasmopara viticola*- and *Botrytis cinerea*-based fungal infections.

## 1. Introduction

Grapes, one of the most valuable cash crops, play an important role in the global economy. In 2020, the global production of grapes was 78 million tonnes from 6.95 million hectares, and the total production value was over USD 80 billion [1], with ~6 million tons of grapes produced in the U.S. [2]. Cultivated grapes are sold as table grapes and processed grape products such as jam, wine, vinegar juice, and jelly. Over 50% of grapes are used in wine production, which contributes to over a billion U.S. dollars each year in the U.S. However, fungal diseases have a serious impact on the growth of grapes, which also affects the quality of wine and other products. An estimated 40% reduction in grape production occurs annually because of various fungal diseases, causing significant economic losses [3].

Downy mildew caused by *Plasmopara viticola* is one of the most serious fungal diseases that attack grapevines. Downy mildew is a native pathogen to North America and caused serious damage to European vineyards in the late 1800s [4]. *Plasmopara viticola* invades leaves, shoots, and young berries under warm and moist conditions, allowing the pathogen to take nutrients from these parts to produce sporangia, causing larger infections [4,5]. Yellow spots with white downy mold occur on the surfaces of infected leaves, and the spots turn brown and eventually necrotic [6]. Necrotic areas on leaves caused by downy mildew largely affect the photosynthesis of the grapevine and reduce the formation of glucose provided by photosynthesis, which hinders grape growth and causes a reduction of berries. Young shoots and berries are also vulnerable to downy mildew, which causes young shoots to become twisted and decreases the translocation of water and organic nutrients, slowing growth [6]. This infection also causes berries to dry out and fall, resulting in significant losses [6]. When there is abundant rainfall in warm seasons, the pathogen easily invades grapevines and reproduces more sporangia, which can be carried by wind or rain to infect surrounding grapevines [6,7]. Although downy mildew is devastating, QoI fungicides have thus far been able to manage the disease effectively [6,7,8]; however, a key mutation in the target results in resistance to several of these fungicides.

Gray mold caused by *Botrytis cinerea* is another fungal disease that causes serious destruction to grapevines. Gray mold is also one of the “Top 10 fungal plant pathogens” in the survey established by *Molecular Plant Pathology* because this fungal disease has a wide host range and can invade a host plant in all stages, from seedling to maturity [9]. The berries of grapevine are the most susceptible when an infection occurs at moderate temperatures and high humidity [10]. When the berries are infected, a reddish-brown and watery decay can be observed from the pedicel to the stylar end [10]. Infected regions also provide favorable conditions for a secondary inoculum, which will generate more sporangia and infect other berries nearby [10]. Infected berries finally dry out, resulting in significant economic losses. *Botrytis cinerea* also invades leaves, flowers, and shoots, causing similar brown lesions on plant parts [10]. QoI fungicides are commonly used for chemical control of *Botrytis cinerea* [10], and resistance threatens the effectiveness of several of these fungicides.

### Literature Review

The application of fungicide Is a chemical control that targets specific molecules such as amino acids to block fungal metabolism, restricting fungal reproduction [11]. The binding target of QoI fungicides is cytochrome b, a protein within the cytochrome bc_1_ complex in *Plasmopara viticola* and *Botrytis cinerea* that plays a significant role in respiratory function [12]. When QoI fungicides bind cytochrome b, the ubiquinol oxidase substrate is unable to transfer electrons between cytochrome b and cytochrome c_1_, interrupting and inhibiting the production of adenosine triphosphate (ATP) [8,12]. This shortage of ATP interrupts the propagation of the pathogen, meaning downy mildew treated by QoI fungicides cannot infect other parts of the grapevine. However, since QoI fungicides are single-site fungicides (specifically binding to cytochrome b), downy mildew and gray mold will develop fungicide resistance after continuous usage of QoI fungicides [12,13,14]. Because of fungicide resistance development, the Fungicide Resistance Action Committee (FRAC) labeled QoI fungicides as high-risk. The G143A (glycine to alanine) mutation was the main known mutation that reduced the efficacy of QoI fungicides toward grape downy mildew because the mutation of this target site weakens the binding affinity between protein and the fungicides [12,15,16]. While developing new types of fungicides can resolve this issue, it takes a significant amount of time and resources to identify alternative active sites and undergo the rigorous approval processes. In the meantime, an effective, economical, and viable management strategy may include using combinations of fungicides. These combine one or more high-risk fungicides with one or more low-risk fungicides from those currently used [15]. With this strategy, QoI fungicides can be combined with another low-risk fungicide, providing the mixture with multiple binding targets and thus increasing the effectiveness against mutation [15].

Currently, only a few studies exist recommending fungicide combinations targeting *Plasmopara viticola* and *Botrytis cinerea* resistance, as summarized in Table 1. There is a gap in knowledge on what fungicide combinations should be selected to treat crops that have already developed resistance and also on what combinations would be most logical to use to prevent developing resistance. The approach we have taken in this study is to evaluate how fungicides interact with WT and mutated versions of the target protein(s) at a molecular level to screen the highest-binding fungicides, which in turn will be recommended for field testing.

Select studies on fungicide combinations are shown in Table 1. A key limitation is that, since the studies were experimentally based, only a limited number of fungicides could be tested. Moreover, the long-term efficacy of these combinations is yet to be determined. A key advantage of in silico-based methods is their ability to screen a large number of fungicides simultaneously based on rational molecular-level information and select those with the highest promise for field testing. According to the existing literature, no studies have provided guidance for the selection of fungicide combinations based on molecular structures and their affinity for the target active site.

The significance of this work is that, as of now, there is only limited information for researchers to follow on selecting fungicides for *Plasmopara viticola* and *Botrytis cinerea* once resistance to existing fungicides is suspected. Also, there is no objective way to suggest fungicide combinations to reduce the development of resistance or treat crops that have already developed fungicide resistance. This study aims to provide a thermodynamic-coupled machine learning strategy to identify and select antifungal agents from QoIs (high-risk group) combined with low-risk fungicides to form fungicide combination(s) that can mitigate fungicide resistance. This approach is based on docking selected fungicides from QoIs and low-risk fungicides with a homology model of cytochrome b to identify the fungicides with the highest affinity and evaluating the screened fungicides using QSAR models with machine learning statistical methods.

## 2. Materials and Methods

### 2.1. Protein Structure and Ligand Structure Preparation

A homology model in PDB format of *Plasmopara viticola* (GenBank: DQ209286.1) was created using cytochrome b from plant mitochondrial complex III2 from *Viga radiata* (PDB: 7JRG. 1 .C) as a template on the SWISS-MODEL server [11,22,23,24]. The quality of this homology model was evaluated using the programs ERRAT and PROVE on the SAVES v6.0 server [25,26]. The homology model contained G143 and F129, which were forms of WT cytochrome b. This model was mutated into G143A-mutated versions by Maestro Schrödinger. The canonical SMILES formulas of ligands were obtained from ZINC15 or PubChem (all 2D ligand structures are provided in Appendix A Appendix A), and the 3D structures of these ligands were generated in PDB format using the online SMILES translator [27,28]. All the protein and ligand structures were prepared for docking using the Protein Preparation Wizard, which added missing hydrogens, corrected bond orders, fixed missing segments, and minimized the structure under the Optimized Potentials for Liquid Simulations 3 (OPLS3) force field [29].

A homology model was built and validated for *Botrytis cinerea* using methods analogous to those developed for *Plasmopara viticola.*

### 2.2. Molecular Docking

Schrödinger Glide was used for the docking of the ligands on the protein. The grid box was centered around the original active sites (G143 and F129; coordinates X—195.53, Y—213.29, Z—176.3) or mutated active sites (G143A: coordinates X—192.6, Y—212.54, Z —171.55) and the size of the grid box was 44 × 46 × 56 Å. Glide docking scores between cytochrome b and the 27 ligands (ubiquinol and compounds 1–26 in Appendix A) were generated using Schrödinger Glide XP mode with default settings in three replicates, and the highest binding scores were used for binding affinity analysis. The ligand-protein interactions were analyzed using a ligand interaction diagram.

### 2.3. AutoQSAR Model Analysis

In order to evaluate the predictions made by the docking, the Schrödinger automated quantitative structure–activity relationship (AutoQSAR) model with a machine-learning approach that is a subset of Artificial Intelligence (AI) was used. The AutoQSAR model is a machine-learning approach that builds numerical models with minimal inputs to interpret the relationship and make predictions between the bioactivity and chemical properties of ligands [30]. In this case, the binding affinity was used as the input variable. Numerical models were developed using multiple linear regression (MLR), partial least-squares regression (PLS), kernel-based partial least-squares regression (KPLS), and principal components regression (PCR) based on the given ligands’ fingerprints, including linear, radial, dendritic, and milprint2D, or descriptors [30]. The AutoQSAR split the selected ligands into a 75% training set and a 25% test set for *Plasmopara viticola* and *Botrytis cinerea* [30]. This model generated a scatter plot that showed the correlation between observed and predicted binding affinity. The accuracy of this model was evaluated by an external validation data set.

More information on the fungicides that were used in the in silico study is given in Table 2. The structural information of the compounds used in both in silico and QSAR studies is given in Appendix A.

## 3. Results

### 3.1. Building of the Homology Models for Plasmopara viticola Cytochrome b

The homology model of *Plasmopara viticola* cytochrome b included the regions between residues 79 and 295 found by BLAST. The sequence identity was 69.59%, and the sequence similarity was 0.53 [24]. The Quaternary Structure Quality Estimate (QSQE) was 0.81 (with 0.7 as acceptable), indicating a high level of reliability. The Global Model Quality Estimate (GMQE) was 0.89, meaning the homology model had over half of the target sequence coverage [24]. The ERRAT value for the homology model was 93.5, which meant that the homology model had acceptable nonbonded atomic interactions [25]. Based on the PROCHECK report, 92.9% of residues (171 out of 217) were located in the most favored regions, 6.5% (12 out of 217) were located in the additional allowed regions, 0% were located in the generously allowed regions, and 0.5% (1 out of 217) were located in the disallowed regions [31]. The residue score provided by PROCHECK was 99.5%, which indicated that the conformation of the homology model of *Plasmopara viticola* cytochrome b was stable [32]. Based on favorable scores, this homology model was used for in silico studies.

### 3.2. Identification of the Active Site for Plasmopara viticola

During the initial docking, the homology model was divided into top (covering residues ARG79-TRY94, MET124-PHE180, and PHE245-MET295) and bottom sections (covering residues ILE95-PHE121 and SER181-ILE244). To identify the docking site in cytochrome b, two conserved regions around the center of the protein were picked from the top and bottom parts based on the existing literature [33]. Initial docking results and the site map shown in Figure 1 revealed the strong binding of probe molecules to the top region (covering residues ARG79-TRY94, ILE122-PHE180, and PHE245-MET295) of cytochrome b. Since the top region showed stronger binding and included the residue where the antifungal-resistant mutation occurred, this region was used for docking analyses in all subsequent steps. The key interactions of ubiquinol at the binding site are given in Figure 1. Cytochrome b had strong hydrophobic interactions with ubiquinol, which was also predominant between cytochrome b and the fungicides tested (Figure 1). Hydrogen bonding was also observed with ARG178 for WT cytochrome b. Ubiquinol formed strong hydrogen bonds with MET295 of the G143A-mutated type.

### 3.3. Fungicide Binding Behavior on Plasmopara viticola Cytochrome b

Since the focus of this study was to identify fungicides that were effective against WT and the G143A mutation of cytochrome b, a set of known antifungal agents were docked onto WT and G143A-mutated types of *Plasmopara viticola* cytochrome b. Here, the G143A mutation was specifically selected since it was reported to be most significant for antifungal resistance [33].

### 3.4. Mutation-Specific Observations

In order to reveal any specific interactions of fungicides with a particular mutation, the statistical analysis focused on each of the individual versions of *Plasmopara viticola* cytochrome b. This type of analysis will be helpful in identifying the best possible fungicide(s) if the mutation is known.

#### 3.4.1. Fungicide Recommendations for WT

In the case of WT, ubiquinol showed a strong binding affinity to WT cytochrome b (Figure 2), as expected. Mandestrobin, fenaminstrobin, dimoxystrobin, fenamidone, famoxadone, and ametoctradin had a stronger affinity than other high-risk fungicides, meaning they were effective agents for WT cytochrome b. Metominostrobin and thiram had higher affinities than the other fungicides, indicating that they were also effective against cytochrome b. Pyraoxystrobin, pyrametostrobin, pyraclostrobin, flufenoxystrobin, coumoxystrobin, picoxystrobin, triclopyricarb, orysastrobin, fluoxastrobin, and metyltetraprole did not bind to WT cytochrome b, and thus extensive usage of these fungicides has a high propensity to develop resistance. Azoxystrobin and ametoctradin are two fungicides known to be resistant to *Plasmopara viticola* [34,35]. Ametoctradin showed a somewhat strong affinity toward the WT version, although the docking score toward WT cytochrome b was lower than fenamidone, dimoxystrobin, fenaminstrobin, famoxadone, and mandestrobin. Azoxystrobin had a poor docking score, indicating that it may not be effective against cytochrome b inhibition. Among low-risk fungicides, i.e., fungicides having more than one MOA, thiram showed a strong affinity toward WT cytochrome b. Other low-risk fungicides such as captan, folpet, ferbam, and zineb did not bind tightly to WT cytochrome b, which meant that these low-risk fungicides were not recommended because of their high potential susceptibility to resistance.

#### 3.4.2. Fungicide Recommendations for the G143A Mutation

Ubiquinol as a native substrate also showed a strong affinity for the G143A mutation of cytochrome b (Figure 3). Mandestrobin, fenaminstrobin, dimoxystrobin, fenamidone, famoxadone, and ametoctradin, which showed strong affinity toward WT cytochrome b and were effective agents against G143A-mutated cytochrome b. Coumoxystrobin, flufenoxystrobin, pyribencarb, and metominostrobin did not show a high binding affinity to WT cytochrome b, but they were effective fungicides when the G143A mutation occurred, meaning the interaction between the G143A-mutated version and those ligands was stronger than WT cytochrome b. Pyraoxystrobin, pyrametostrobin, pyraclostrobin, flufenoxystrobin, enoxastrobin, picoxystrobin, triclopyricarb, orysastrobin, fluoxastrobin, and metyltetraprole did not bind to G143A-mutated cytochrome b, indicating that these high-risk fungicides were not preferred for G143A-mutated cytochrome b. Low-risk fungicides folpet and thiram showed higher binding affinities than ferbam, zineb, mancozeb, and captan, which meant folpet and thiram would be more effective fungicides for G143A-mutated cytochrome b. As a resistant fungicide, azoxystrobin did not bind to either WT or G143A-mutated cytochrome b as expected.

A common recommendation is to use fungicide combinations that consist of different MOAs, i.e., combining one MOA with others in a fungicide rotation program. Because of their ability to tackle mutation, fenamidone, famoxadone, mandestrobin, dimoxystrobin, fenaminstrobin, ametoctradin, and thiram have been identified as suitable candidates for a rotational program targeting *Plasmopara viticola*.

The top conformation interactions of the highest affinity fungicides with G143A-mutated versions are given in Figure 4. Dimoxystrobin showed strong hydrophobic and hydrogen bonding interactions with MET125 of G143A-mutated cytochrome b. It was evident that the primary interactions between fungicides and cytochrome b were hydrophobic, which agreed with the predominantly hydrophobic nature of cytochrome b proteins [36,37]. Figure 4 shows dimoxystrobin forming strong hydrophobic interactions with the ILE122-ILE147 and PHE275-MET295 regions in the G143A-mutated versions. Of the low-risk fungicides, thiram showed strong hydrophobic interactions with cytochrome b.

Based on the binding analysis (Figure 5), the pocket located on the top region of cytochrome b that contained the residues F129 and G143 seemed to be an important binding position when targeting *Plasmopara viticola* inhibition. Ametoctradin, famoxadone, fenamidone, fenaminstrobin, mandestrobin, dimoxystrobin, metominostrobin, and thiram tended to bind to this pocket, including the native substrate ubiquinol.

### 3.5. Fungicide Binding Behavior on Botrytis cinerea Cytochrome b

To further verify binding affinities, an additional set of docking simulations were performed, this time using a grid box covering the ubiquinol binding site and the specific residues G143 and F129 on cytochrome b of *Botrytis cinerea*. For this analysis, the same 26 fungicides (Compounds 1–26 in Appendix A), including resistant, high-risk, and low-risk, were selected. *Botrytis cinerea* was used because *Plasmopara viticola* was an obligate parasite and experimental validations could only be performed under field conditions, whereas *Botrytis cinerea* validations could easily be carried out in a laboratory setting.

### 3.6. Mutation-Specific Observations

#### 3.6.1. Fungicide Recommendations for WT

It was observed that pyraoxystrobin, mandestrobin, enoxastrobin, and pyribencarb had higher binding affinity than ubiquinol to WT cytochrome b, indicating their potential superiority as effective fungicides via inhibition of cytochrome b of *Botrytis cinerea* (Figure 6). Fenaminstrobin, pyraclostrobin, dimoxystrobin, famoxadone, metominstrobin, pyrametostrobin, flufenoxystrobin, picoxystrobin, folpet, ametoctradin, fenamidone, captan, and triclopyricarb had higher affinities to WT cytochrome b than the other fungicides. Azoxystrobin, as an identified resistant fungicide, did not bind to WT cytochrome b. Coumoxystrobin, fluoxastrobin, metyltertrapole, and orysastrobin did not bind to WT cytochrome b, indicating their high likelihood of succumbing to resistance. Moreover, zineb and ferbam were low-risk fungicides that showed weaker binding affinity than captan, thiram, and folpet, which meant zineb and ferbam were not likely to be effective fungicides against cytochrome b of *Botrytis cinerea*.

#### 3.6.2. Fungicide Recommendations for the G143A Mutation

Fungicides famoxadone, mandestrobin, pyribencarb, picoxystrobin, metominostrobin, fenamidone, pyraoxystrobin, and thiram showed a stronger affinity to G143A-mutated cytochrome b of *Botrytis cinerea* than ubiquinol, indicating their superior ability to withstand the resistance caused by the G143A mutation of cytochrome b of *Botrytis cinerea* (Figure 7). Enoxastrobin, fenaminstrobin, flufenoxystrobin, and pyrametostrobin also showed a strong affinity toward WT cytochrome b but did not bind or bound weakly to G143A-mutated cytochrome b, meaning these fungicides may not be effective if the G143A mutation occurred. Coumoxystrobin, fluoxastrobin, metyltertrapole, and orysastrobin did not bind to G143A cytochrome b. Azoxystrobin, as an identified resistant fungicide, did not bind to G143A cytochrome b. Ferbam and zineb had weaker binding affinity, while captan and folpet did not bind to G143A cytochrome b, indicating that these four low-risk fungicides were not likely effective against G143A-mutated cytochrome b of *Botrytis cinerea*.

Among the high-risk fungicides, mansestrobin, pyribencarb, fenamidone, famoxadone, and ametoctradin were effective fungicides against WT and G143A-mutated cytochrome b. Thiram showed a strong affinity for WT and G143A-mutated cytochrome b among all the low-risk fungicides. Azoxystrobin did not show stable binding with WT and the G143A mutation, which was expected since it was resistant to cytochrome b. Because of their ability to tackle mutation, mansestrobin, pyribencarb, fenamidone, famoxadone, ametoctradin, and thiram were identified as suitable candidates for a rotational program targeting *Botrytis cinerea*.

An analysis of the binding behavior of ametoctradin, pyraoxystrobin, mandestrobin, enoxastrobin, and pyribencarb in the vicinity of the G143 and F129 residues of WT cytochrome b of *Botrytis cinerea* indicated that they all bound close to the two residues (Figure 8). For the G143A mutation, picoxystrobin, metominostrobin, pyribencarb, famoxadone, and mandestrobin bound to the same site as WT cytochrome b. This position was also the binding site for ubiquinol on both WT and G143A cytochrome b, indicating that this site is crucial when determining effective QoIs targeting *Botrytis cinerea* cytochrome b inhibition.

An analysis of the interactions (Figure 9) of cytochrome b of WT *Botrytis cinerea* with ubiquinol and pyraoxystrobin indicated that hydrophobic bonding was the primary interaction that occurred. There was also hydrogen bonding with PHE164 and ARG178 for ubiquinol and GLU273 for pyraoxystrobin. For G143A-mutated cytochrome b, hydrophobic bonding still played a major role in pyribencarb. The interaction with the residue F129 was hydrophobic regardless of the ligand. However, the interaction of ligands with the residue G143 was not apparent in WT cytochrome b; however, pyribencarb showed hydrophobic bonding once the G143A mutation occurred.

### 3.7. AutoQSAR Model Evaluation

#### Application of AutoQSR to Predict Fungicides for *Botrytis cinerea*

Training Data without a Validation Set

An initial training set was developed using 16 QoI and 18 non-QoI fungicides. The top five QSAR models and their performance parameters generated for *Botrytis cinerea* are depicted in Appendix A, and these were pls_19, kpls_radial_19, kpls_dendritic_19, kpls_desc_19, and kpls_linear_19. Based on the scoring functions, the best model was pls_19, which was generated by partial least squares regression (PLS) using the 19th split of the learning set into a test and training set (34 ligands) without a validation set. This model had a standard deviation (S.D.) of 2.1184, a R^2^ of 0.6378, a root mean square error (RMSE) of 2.1419, a Q^2^ of 0.6172, and a ranking score of 0.5892. Binding affinity, Y(Obs), and predicted affinity, Y(Pred), from the QSAR model of all selected ligands are shown in Figure 10a (under DATASET) and Appendix A, with 75% of the ligands belonging to the training set and 25% of the ligands to the test set for *Botrytis cinerea*. The five scatter plots in Figure 10 (b) and Appendix A were generated based on Y(Obs) and Y(Pred). The results indicated that about 50% of training sets were close to the regression line.

In order to evaluate whether the predictions could be improved, it was decided to refine the models by systematically removing outliers that were chemically distinct from the ones that functioned as QoIs and/or when the difference between actual and predicted affinities was larger than 3 kCal/mol.

Iteration #1

The new external validation set that included 19 ligands (Appendix A) was used to estimate the prediction accuracy of the QSAR model made by the top five numeric models listed in Appendix A. The R^2^ value of the best-fit line was 0.06, meaning the QSAR model was not able to predict with an acceptable level of accuracy for the given validation set. Moreover, some ligands in Figure 10c fell outside the applicability domain of the QSAR model, which would decrease the prediction accuracy of the QSAR model. Visual inspection of Figure 10c showed that metominostrobin, azaconazole, dithianon, and picarbutrazox were outliers, indicating these ligands were possibly unsuitable for the validation set for the built model. Both azaconazole and picarbutrazox had multiple heterocyclic nitrogen atoms (Appendix A). Dithianon was the only ligand that contained heterocyclic dual sulfur atoms among the 19 ligands. Metominostrobin, azaconazole, dithianon, and picarbutrazox had an oxygen-containing aromatic ring, and there were at least two oxygen atoms in each ligand. The structural features mentioned above might be the reason these four ligands were not suitable for the validation set. To improve the prediction accuracy of the QSAR model, these ligands were considered for removal in the next iteration.

Iteration #2

After removing metominostrobin, azaconazole, dithianon, and picarbutrazox, 15 ligands were next considered in the validation set (Appendix A). The R^2^ value of the best-fit line increased from 0.06 to 0.26, indicating the four outliers might be potential factors affecting the prediction accuracy of the QSAR model. Both furametpyr and iprodione had chlorine in their chemical structure, which was similar to azaconazole, which was removed in first iteration (Appendix A). Furametpyr, iprodione, and penthiopyrad had heterocyclic dual nitrogen atoms. Diethofencarb had a similar structure to metominostrobin, which was removed in the first iteration. Based on the visual inspection of Appendix A and similar chemical structures, furametpyr, iprodione, penthiopyrad, and diethofencarb were considered potential outliers.

Iteration #3

After removing the eight outliers (metominostrobin, azaconazole, dithianon, picarbutrazox, furametpyr, iprodione, penthiopyrad, and diethofencarb) in this iteration, the R^2^ value in Appendix A of the best-fit line increased from 0.26 to 0.53. The rest of the compounds in Appendix A were more acceptable as an external validation set, meaning that they were expected to generate better predictions. The top predictions that would withstand G143A-mutated cytochrome b of *Botrytis cinerea* were fenpropidin (an amine), fenoxanil (a melanin biosynthesis inhibitor dehydratase), isoflucypram (a succinate dehydrogenase inhibitor), and ametoctradin (a QoI). Although oxathiapiprolin and triazoxide had a high predicted binding affinity, their original affinity was low, so they were not acceptable. Chlorine and multiple heterocyclic nitrogen were considered similar to the outliers’ structures.

### 3.8. Training Data with a Validation Set

In this case, two QoI fungicides, picoxystrobin and pyribencarb, were assigned as a validation set for the QSAR models. This resulted in 32 ligands in the training set for the QSAR model (Figure 11a and Appendix A). The ranking scores for the top five QSAR models with a validation set (Appendix A) were higher than the model without the validation set, meaning that the test set predictions of a model with a validation set might be more accurate. The top QSAR models shown for *Botrytis cinerea* in Appendix A were kpls_molprint2D_39, kpls_radial_8, kpls_linear_30, kpls_dendritic_30, and kpls_dendritic_39. The best model was kpls_molprint2D_39, which was generated by kernel partial least squares regression (KPLS) with molprint2D fingerprint, using the 39th split of the learning set into a test and training set (32 ligands) with the validation set (2 ligands). This model had an S.D. of 1.8732, a R^2^ of 0.7081, an RMSE of 1.8919, a Q^2^ of 0.6860, and a ranking score of 0.6582. As seen in the plots in Figure 11b and Appendix A, training sets were closer to the regression line (Figure 10b and Appendix A), indicating the better prediction ability of the model.

Similar to the procedure that was adopted in the previous run, several iterations were conducted while removing chemically distinct outliers.

Iteration #1

In Appendix A, the external validation set had the same 19 ligands to estimate the prediction accuracy of the QSAR model. The R^2^ value of the best-fit line was 0.05, meaning the QSAR model did not perform well for the given validation set. Visual inspection of Supplementary Figure 11c showed that penthiopyrad, isoflucypram, picarbutrazox, and metominostrobin fell outside the applicability domain of the QSAR model, which would decrease the prediction accuracy of the model. Moreover, penthiopyrad and isoflucypram had fluorine in their chemical structures (Appendix A). Penthiopyrad, isoflucypram, and picarbutrazox contained multiple heterocyclic nitrogen atoms. Penthiopyrad, isoflucypram, and metominostrobin had nitrogen-hydrogen structures. To improve the prediction accuracy of the QSAR model, these ligands were removed in the next iteration.

Iteration #2

After removing penthiopyrad, isoflucypram, picarbutrazox, and metominostrobin in Appendix A, the R^2^ value of the best-fit line slightly increased. A visual inspection of Appendix A suggested the presence of several outliers (oxathiapiprolin, azaconazole, flusulfamide, diethofencarb, and dithianon) that would need to be removed to improve the QSAR model. Oxathiapiprolin and flusulfamide had fluorine in their chemical structures (Appendix A). Azaconazole and flusulfamide have chlorine in their chemical structures. Diethofencarb had a similar structure to metominostrobin. Only dithianon contained heterocyclic dual sulfur atoms.

Iteration #3

To further improve the QSAR model, outliers including oxathiapiprolin, azaconazole, flusulfamide, diethofencarb, and dithianon were removed from Appendix A. The R^2^ value of the best-fit line significantly increased from 0.08 to 0.67, indicating those ligands had structural properties that were not predictable using the models (Appendix A). The top predictions that would withstand G143A-mutated cytochrome b of *Botrytis cinerea* were fenoxanil (a melanin biosynthesis inhibitor dehydratase), fenpropidin (an amine), iprodione (a dicarboximde), tebufloquin (a 4-quinolyl-acetate), and ametoctradin (a QoI). Furametpyr and triazoxide had high predicted affinity, but the difference between their original affinity and predicted affinity was higher than the other four ligands. It should be noted that the model was not able to make accurate predictions when ligands had chlorine, fluorine, and heterocyclic nitrogen atoms in their chemical structures.

#### Application of AutoQSR to Predict Fungicides for *Plasmopara viticola*

Training Data without a Validation Set

Here, an initial training set was developed using 16 QoI and 20 non-QoI fungicides. The top five QSAR models for *Plasmopara viticola* were kpls_desc_2, kpls_radial_24, kpls_linear_22, kpls_radial_22, and pls_2 (Appendix A). The best model was kpls_desc_2, generated by kernel partial least squares regression (KPLS) with a desc fingerprint, using the 12th split of the learning set into a test and training set (36 ligands) without a validation set. This model had an S.D. of 1.7498, a R^2^ of 0.7032, an RMSE of 1.615, a Q^2^ of 0.7350, and a ranking score of 0.7116. In Figure 12a and Appendix A, 75% of the ligands occupied the training set and 25% occupied the test set. The scatter plots in Figure 12b and Appendix A indicate that the first four models’ training sets correlated closely with the test sets.

Here, in order to evaluate whether the predictions could be improved, it was decided to refine the models by systematically removing outliers that were chemically distinct from the ones that functioned as QoIs and/or when the difference between actual and predicted affinities was larger than 3 kCal/mol, as previously performed.

Iteration #1

In this iteration, the external validation set had 17 ligands to estimate the prediction accuracy of the QSAR model (Appendix A). The R^2^ value of the best-fit line was not impressive (Figure 12c). Visual inspection of Figure 12c indicated two apparent ligands, fluindapyr and picarbutrazox, falling outside the applicability domain of the QSAR model. Triazoxide, polyoxin, dithianon, and dimoxystrobin also deviated significantly from the regression line. Both fluindapyr and triazoxide had chlorine in their chemical structures (Appendix A). Dithianon was the special ligand that had heterocyclic dual sulfur atoms in aromatic rings. Picarbutrazox and dimoxystrobin had a similar structure. Triazoxide and polyoxin showed oxygen with a negative charge. To improve the prediction accuracy of the QSAR model, these ligands were removed in the next iteration.

Iteration #2

After the first iteration, 6 ligands were removed, and 11 ligands remained in Appendix A. Although the R^2^ value of the best-fit line increased from 0 to 0.25, there were still some outliers, such as furametpyr, fenpropidin, and tebufloquin, shown in Appendix A. Furametpyr had chlorine that was similar to triazoxide in Appendix A. Tebufloquin had fluorine that was similar to fluindapyr. All three ligands (furametpyr, fenpropidin, and tebufloquin) had a similar carbon structure, as shown in Appendix A. To further improve accuracy, these ligands were removed in the next iteration.

Iteration #3

After removing the outliers mentioned in the second iteration in Appendix A, the R^2^ value of the best-fit line (Appendix A) became 0.63, with an acceptable prediction accuracy. The top predictions that would withstand G143A-mutated cytochrome b of *Plasmopara viticola* were flusulfamide (a benzene-sulfonamide), ametoctradin (a QoI), ethaboxam (a thiazole carboxamide), and famoxadone (a QoI). Isoflucypram and diethofencarb showed high predicted affinity with low original affinity; thus, they were not an appropriate validation set for the QSAR prediction model. Outliers for this QSAR model carried fluorine, chlorine, and oxygen with a charge, and an aromatic ring with sulfur.

Training Data with a Validation Set

In this case, two QoI fungicides, fenaminstrobin and fenamidone, were assigned as the validation set for the QSAR models. Consequently, there were 34 ligands used for the QSAR model (Figure 13a and Appendix A). The ranking scores for the top five QSAR models with a validation set (Appendix A) were higher than for the model without a validation set. The results of the QSAR models from both *Botrytis cinerea* and *Plasmopara viticola* showed that a validation set may provide more accurate prediction models. The top QSAR models shown for *Plasmopara viticola* in Appendix A were kpls_linear_39, kpls_desc_31, kpls_dendritic_31, kpls_linear_2, and kpls_linear_31. The best model was kpls_linear_39, which was generated by kernel partial least squares regression (KPLS) with a linear fingerprint, using the 39th split of the learning set into a test and training set (34 ligands) with a validation set. This model had an S.D. of 1.4315, a R^2^ of 0.7953, an RMSE of 1.4160, a Q^2^ of 0.7624, and a ranking score of 0.7733. As seen in the scatter plots in Figure 13b and Appendix A, the pattern of training sets was similar to the plots in Figure 12b and Appendix A.

Iteration #1

The predicted binding affinities in Appendix A were slightly lower than those in Appendix A. The R^2^ value of the best-fit line was 0.03, as shown in Appendix A. Visual inspection of Figure 13c identified three apparent ligands (fluindapyr, picarbutrazox, and dimoxystrobin) falling outside the applicability domain of the QSAR model, which affected the prediction accuracy of the QSAR model. Dithianon was another ligand that lay further from the regression line, and it was the only ligand that contained an aromatic ring with sulfur (Appendix A). Fluindapyr and picarbutrazox had similar aromatic rings. To improve the prediction accuracy of the QSAR model, these ligands were removed during the next iteration.

Iteration #2

After the first iteration, 4 ligands were removed, and 13 ligands remained in Appendix A. The R^2^ value of the best-fit line shown in Appendix A improved from 0.03 to 0.16 in this iteration (Appendix A). Furametpyr, flusulfamide, tebufloquin, triazoxide, polyoxin, famoxadone, and mandestrobin were outliers based on visual inspection. Furametpyr, flusulfamide, and triazoxide had chlorine. Flusulfamide and tebufloquin had fluorine in their ligand structures (Appendix A). Both triazoxide and polyoxin had oxygen with a charge in their ring structures. Famoxadone and mandestrobin had a similar structure to dimoxystrobin. For further improvement of the model, these ligands were removed.

Iteration #3

In this case, the R^2^ value of the best-fit line shown in Appendix A was 0.64, meaning the prediction accuracy of the QSAR models was acceptable. The top predictions that would withstand G143A-mutated cytochrome b of *Plasmopara viticola* were fenpropidin (an amine), ametoctradin (a QoI), and ethaboxam (a thiazole carboxamide). Isoflucypram, diethofencarb, and fluoxapiprolin were not appropriate selections since their predicted affinities were very different from their original affinities (Appendix A). Outliers for this QSAR model also contained fluorine and chlorine, similar to the QSAR model with a validation set.

## 4. Discussion

The primary purpose of this study was to use in silico simulations to select the highest affinity QoI fungicides for the cytochrome b targets of *Plasmopara viticola* and *Botrytis cinerea*. Based on different in silico simulation methods that consisted of generalized and site-directed ligand impingement methods, docking simulations showed ubiquinol to be the highest affinity ligand for cytochrome b, regardless of the sourced organism. Ubiquinol is bound to cytochrome b primarily via hydrophobic interactions.

In the case of WT cytochrome b of *Plasmopara viticola*, mandestrobin, fenaminstrobin, dimoxystrobin, fenamidone, famoxadone, and ametoctradin bound with the highest affinity and were thus considered effective fungicides. They were also effective agents against G143A-mutated cytochrome b. Coumoxystrobin, flufenoxystrobin, and pyribencarb showed a strongly poor affinity toward WT and the G143A-mutated version, suggesting their susceptibility to potential resistance. As a resistant fungicide, azoxystrobin did not bind to WT and G143A-mutated cytochrome b as expected. Although folpet, a low-risk FRAC code fungicide, showed a reasonable affinity toward G143A-mutated cytochrome b, only thiram had stable and strong affinities toward both WT and G143A-mutated types of cytochrome b among the selected low-risk fungicides. According to the general analysis, from the high- and low-risk groups, mandestrobin, fenaminstrobin, dimoxystrobin, famoxadone, fenamidone, ametoctradin, and thiram emerged as those with the strongest affinity toward *Plasmopara viticola* cytochrome b.

Pyribencarb, mandestrobin, fenamidone, famoxadone, and ametoctradin were effective agents against WT and G143A-mutated cytochrome b of *Botrytis cinerea*. While pyraoxystrobin and metominostrobin had a strong affinity toward WT cytochrome b, their affinity was poor toward mutated cytochrome b. The low-risk fungicides, folpet and captan, had a strong affinity with WT cytochrome b but did not bind to G143A-mutated versions of *Botrytis cinerea* cytochrome b. Thiram showed consistent but moderate affinities.

According to the binding affinity simulation analysis, mandestrobin emerged as the top binder for both *Plasmopara viticola* and *Botrytis cinerea* cytochrome b, regardless of common mutations. Thiram, on the other hand, emerged as a reasonable, low-risk fungicide that worked on WT and the G143A-mutated versions of both fungi. However, the affinity analysis clearly indicated the difficulty of making such broad-spectrum recommendations because of the peculiarities of cytochrome b proteins within different organisms.

Based on a QSAR analysis with an extended array of fungicides, fenpropidin (an amine), fenoxanil (a melanin biosynthesis inhibitor dehydratase), isoflucypram (a succinate dehydrogenase inhibitor), and ametoctradin (a QoI) emerged as effective against G143A-mutated cytochrome b of *Botrytis cinerea*. Moreover, fenoxanil, fenpropidin, iprodione (a dicarboximde), tebufloquin (a 4-quinolyl-acetate), and ametoctradin emerged as high-affinity inhibitors in an analysis with a secondary validation set. The QSAR analysis without a validation set revealed flusulfamide (a benzene-sulfonamide), ametoctradin, ethaboxam (a thiazole carboxamide), and famoxadone (a QoI) emerged as high-affinity fungicides on G143A-mutated cytochrome b of *Plasmopara viticola*. Moreover, fenpropidin, ametoctradin, and ethaboxam showed a strong affinity in the analysis with a secondary validation set. Based on both the docking simulations and QSAR analysis, ametoctradin emerged as a potential high-affinity QoI fungicide toward the G143A-mutated cytochrome b.

## 5. Conclusions

It was observed that mandestrobin, fenaminstrobin, dimoxystrobin, fenamidone, and famoxadone bound to the WT, whereas dimoxystrobin and fenaminstrobin bound to the G143A-mutated cytochrome b of *Plasmopara viticola* with the highest affinity. Although the low-risk fungicide folpet showed reasonable affinity toward G143A-mutated cytochrome b, only thiram had stable and strong affinities toward both WT and G143A-mutated types of cytochrome b among the selected low-risk fungicides. In general, from the high- and low-risk groups, dimoxystrobin, fenaminstrobin, and thiram showed the strongest affinity toward *Plasmopara viticola* cytochrome b. Pyribencarb, mandestrobin, enoxastrobin, and pyraoxystrobin had a high affinity toward WT, whereas famoxadone, mandestrobin, pyribencarb, picoxystrobin, and metominostrobin bound tightly to G143A-mutated cytochrome b of *Botrytis cinerea*. Folpet and captan had a strong affinity with WT cytochrome b but did not bind to G143A-mutated versions of *Botrytis cinerea* cytochrome b. Thiram showed consistent but moderate affinities. Mandestrobin emerged as the top binder for both WT *Plasmopara viticola* and WT *Botrytis cinerea* cytochrome b. Famoxadone appeared to be a versatile binder for G143A-mutated cytochrome b of both *Plasmopara viticola* and *Botrytis cinerea.* Thiram emerged as a reasonable, low-risk non-QoI fungicide that works on WT and G143A-mutated versions of both fungi. QSAR analysis revealed fenpropidin, fenoxanil, and ethaboxam non-QoIs to have a high affinity for G143A-mutated cytochrome b of *Plasmopara viticola* and *Botrytis cinerea*. It is recommended that modeling results be experimentally validated via in vitro and/or plant field studies while attempting to improve the accuracy of the QSAR model using the experimental (validation) data.

## Figures and Tables

**Figure 1 microorganisms-11-01341-f001:**
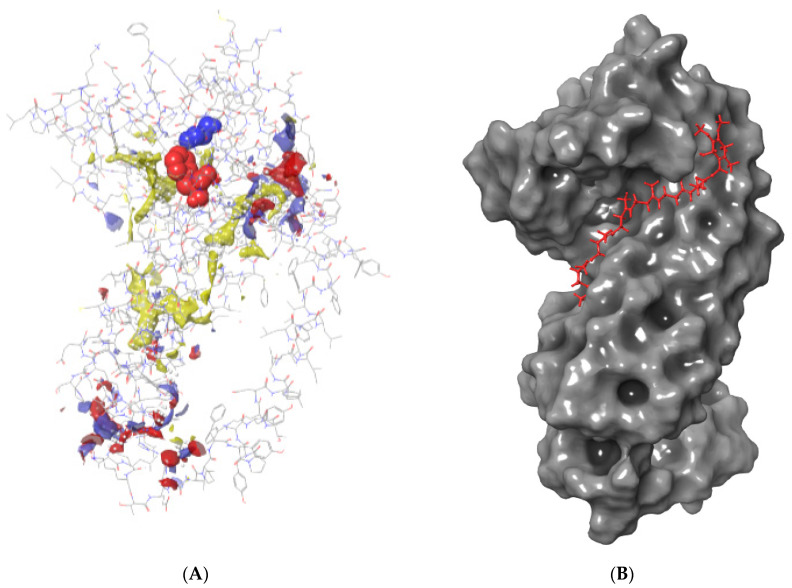
(**A**) Site map output depicting possible binding sites of *Plasmopara viticola* cytochrome b (G143—blue residue; F129—red residue; hydrophobic—yellow areas; hydrogen-bonding acceptor—red areas; hydrogen-bonding donor—blue areas); (**B**) the top orientation of ubiquinol on cytochrome b (ubiquinol—red sticks; WT cytochrome b—gray surface); and (**C**) key interactions of ubiquinol with cytochrome b amino acid residues.

**Figure 2 microorganisms-11-01341-f002:**
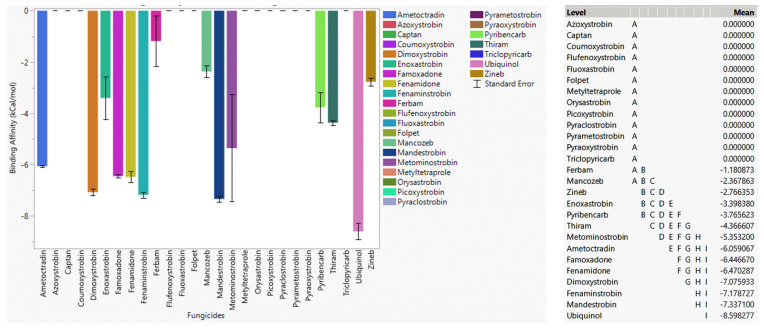
The performance of selected QoI fungicides on WT cytochrome b of *Plasmopara viticola* in a specific grid box. Mean—average binding affinity of three replicates (generated by Glide docking on Maestro Schrödinger) of the corresponding ligand; level—ligands with the same letter are not significantly different (letter level A: ligands with the lowest binding affinity; letter level I: ligands with the highest binding affinity).

**Figure 3 microorganisms-11-01341-f003:**
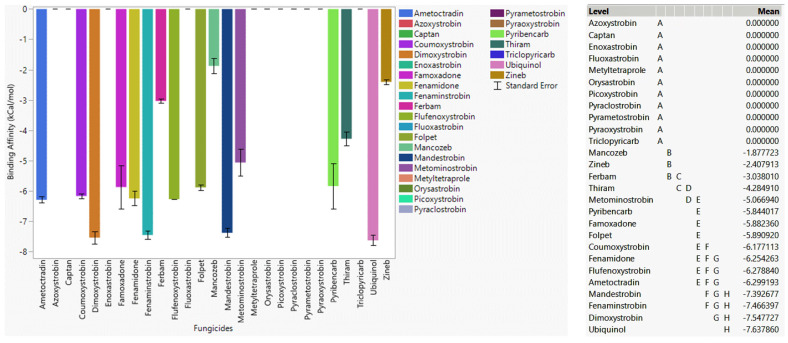
The performance of selected QoI fungicides on G143A-mutated cytochrome b of *Plasmopara viticola* in a specific grid box. Mean—average binding affinity of three replicates (generated by Glide docking on Maestro Schrödinger) of the corresponding ligand; level—ligands with the same letter level are not significantly different (letter level A: ligands with the lowest binding affinity; letter level H: ligands with the highest binding affinity).

**Figure 4 microorganisms-11-01341-f004:**
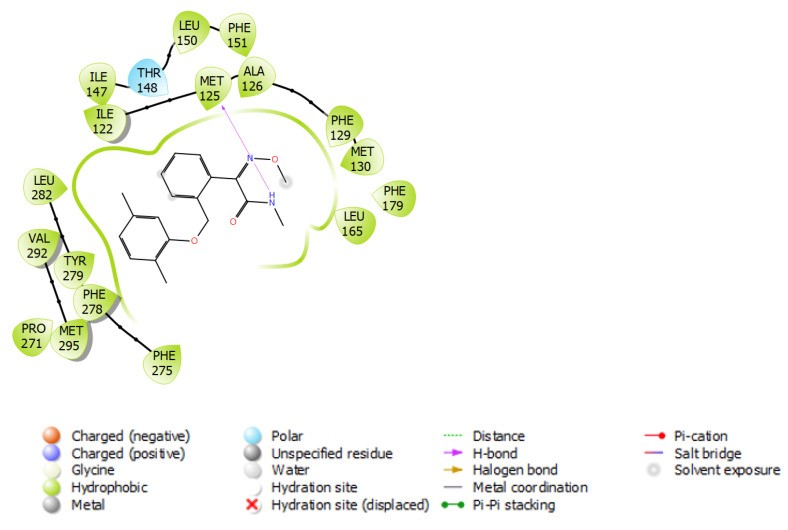
Binding interactions of dimoxystrobin with the G143A-mutated type of cytochrome b.

**Figure 5 microorganisms-11-01341-f005:**
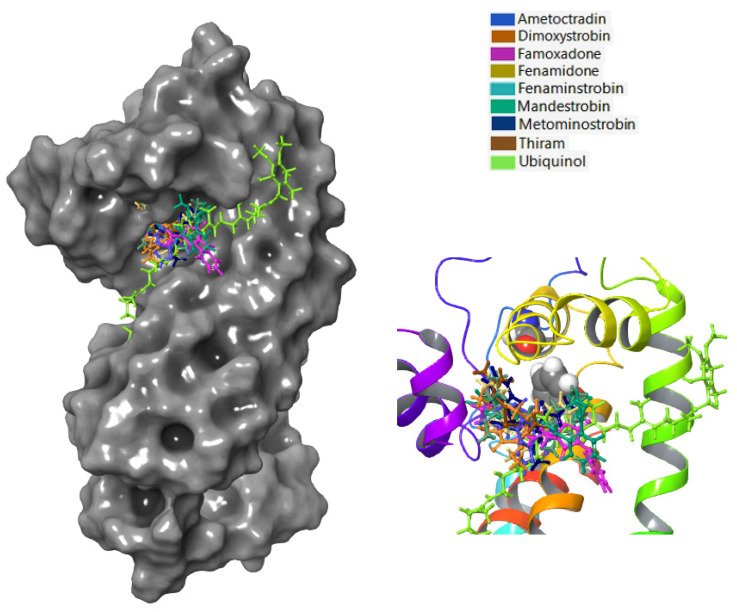
Ametoctradin, Famoxadone, Fenamidone, Fenaminstrobin, Mandestrobin, Dimoxystrobin, Metominostrobin, Thiram, and Ubiquinol are visualized as sticks of different colors bound to WT cytochrome b of *Plasmopara viticola.* Here, WT cytochrome b is represented as a gray surface, while different fungicides are depicted in different colors. The figure to the right represents a close-up of the active site, with the protein represented as a rainbow-colored ribbon and F129 and G143 represented as ball structures.

**Figure 6 microorganisms-11-01341-f006:**
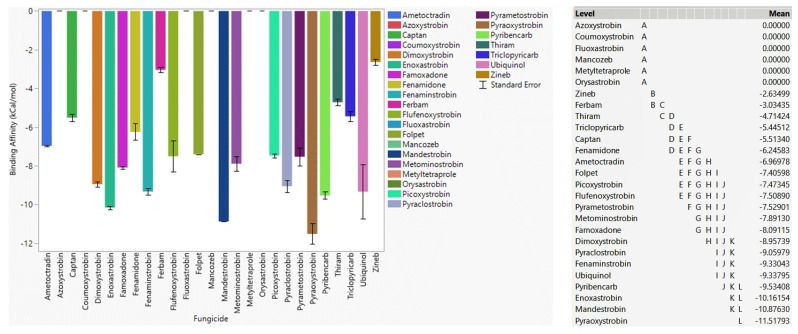
The performance of selected QoI fungicides on WT cytochrome b of *Botrytis cinerea* in a specific grid box. Mean—average binding affinity of three replicates (generated by Glide docking on Maestro Schrödinger) of the corresponding ligand; Level—ligands with the same letter level are not significantly different (letter level A: ligands with the lowest binding affinity; letter level L: ligands with the highest binding affinity).

**Figure 7 microorganisms-11-01341-f007:**
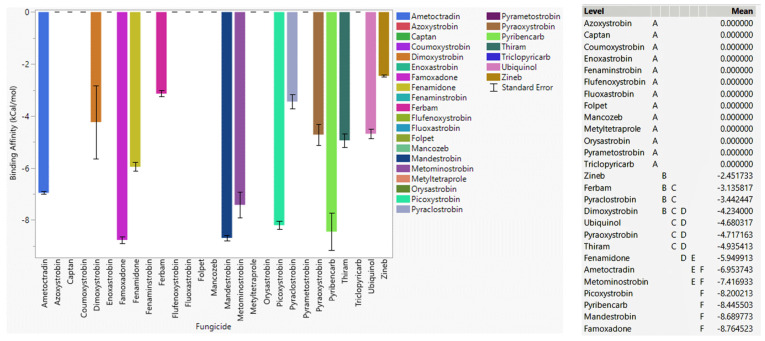
The performance of selected QoI fungicides on G143A cytochrome b of *Botrytis cinerea* in a specific grid box. Mean—average binding affinity of three replicates (generated by Glide docking on Maestro Schrödinger) of the corresponding ligand; Level—ligands with the same letter level are not significantly different (letter level A: ligands with the lowest binding affinity; letter level F: ligands with the highest binding affinity).

**Figure 8 microorganisms-11-01341-f008:**
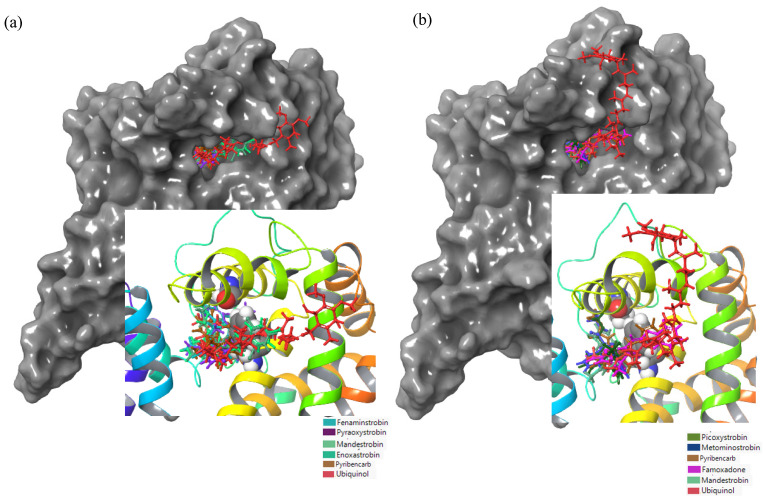
(**a**) Fenamidtrobin, Pyraoxystrobin, Mandestrobin, Enoxastrobin, Pyribencarb, and Ubiquinol with WT cytochrome b and (**b**) Picoxystrobin, Metominostrobin, Pyribencarb, Famoxadone, Mandestrobin, and Ubiquinol are visualized as sticks of different colors bound to G143A cytochrome b of *Botrytis cinerea*. Here, WT and G143A-mutated cytochrome b are represented as gray surfaces while different fungicides are depicted in different colors. The figure to the right represents a close-up of the active site, with the protein represented as a rainbow-colored ribbon and F129 and G143 or G143A represented as ball structures. The colors in the key are for stick models.

**Figure 9 microorganisms-11-01341-f009:**
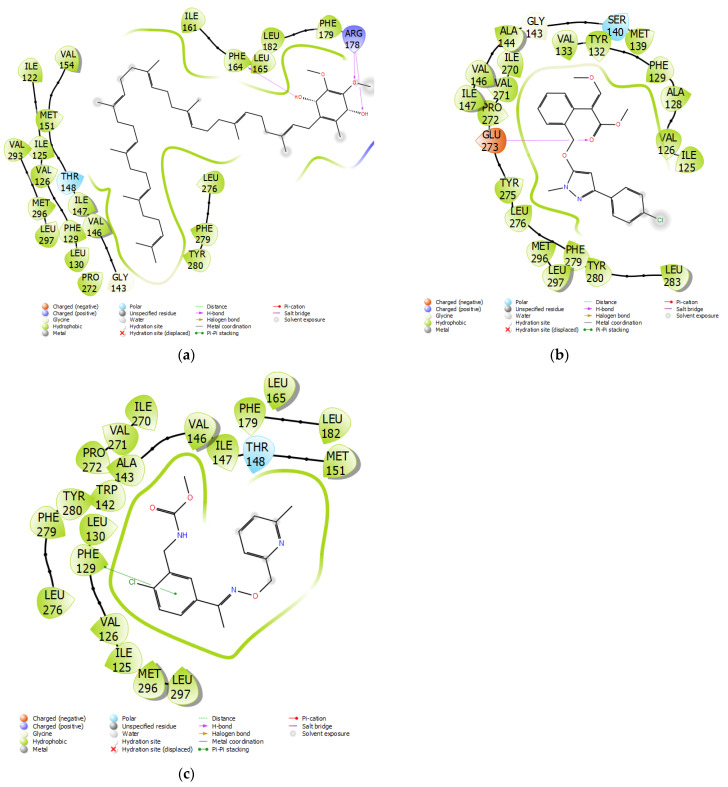
(**a**) Ubiquinol and (**b**) Pyraoxystrobin interactions with WT cytochrome b, and (**c**) Pyribencarb interactions with the G143A mutation of cytochrome b of *Botrytis cinerea*.

**Figure 10 microorganisms-11-01341-f010:**
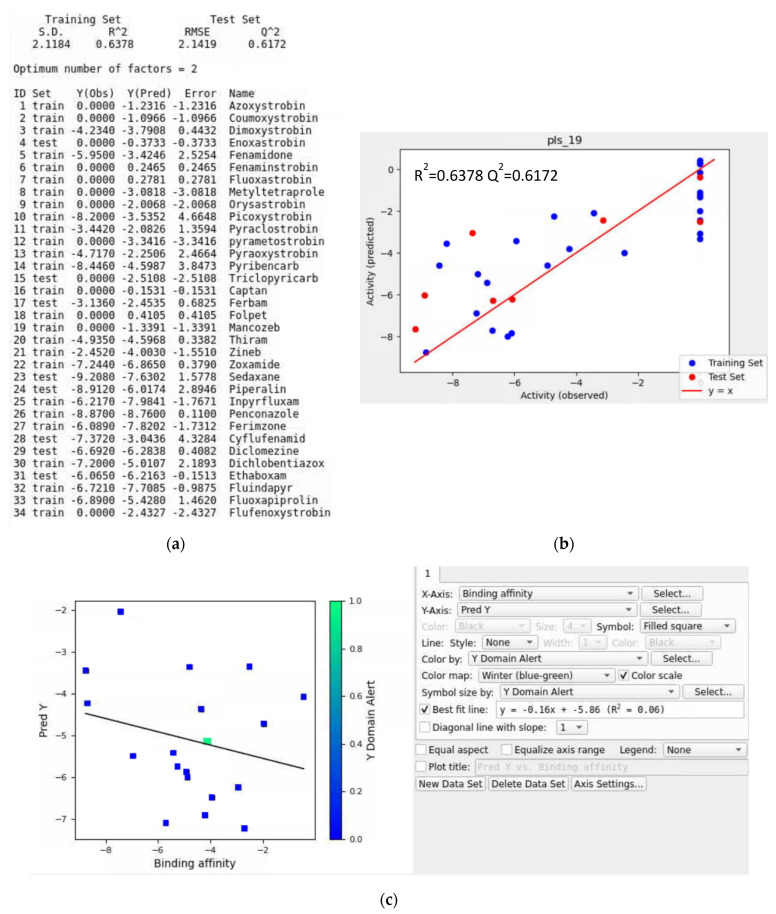
(**a**) Model report for the pls_19 model, (**b**) scatter plot providing the performance for the pls_19 model, and (**c**) scatter plot of the prediction set for the top QSAR models in Appendix A for *Botrytis cinerea* without using a validation set.

**Figure 11 microorganisms-11-01341-f011:**
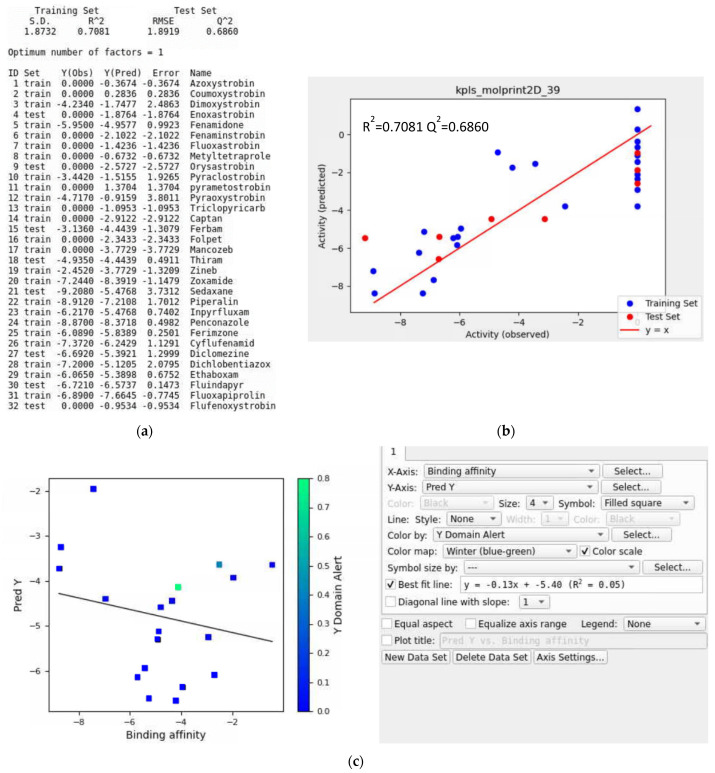
(**a**) Model report for the kpls_radial_39 model, (**b**) scatter plot providing the performance of the kpls_radial_39 model and (**c**) scatter plot of the prediction set for the top QSAR models in Appendix A for *Botrytis cinerea* using a validation set.

**Figure 12 microorganisms-11-01341-f012:**
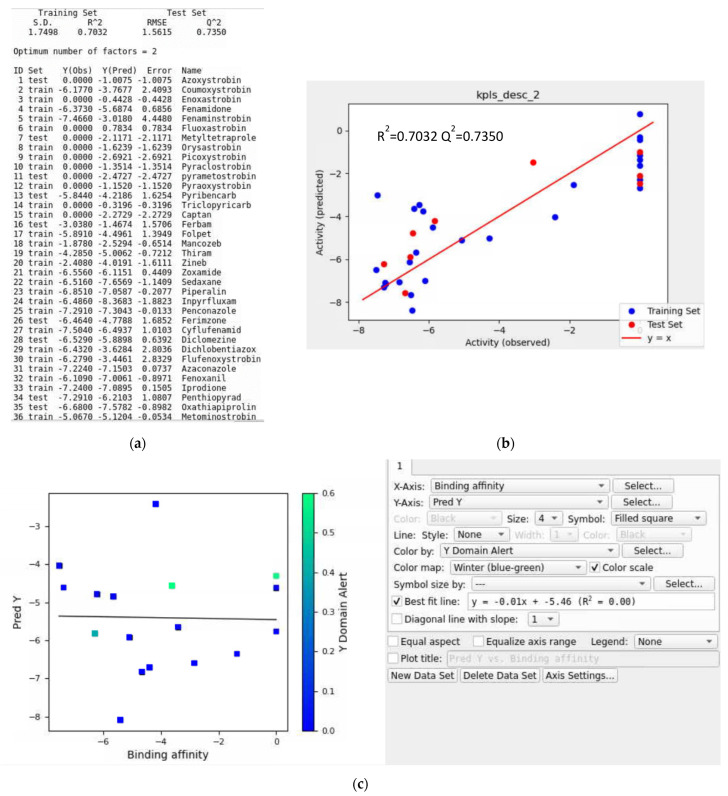
(**a**) Model report for the kpls_desc_2 model, (**b**) scatter plot providing the performance for the kpls_desc_2 model, and (**c**) scatter plot of the prediction set for the top QSAR models in Appendix A for *Plasmopara viticola* without using a validation set.

**Figure 13 microorganisms-11-01341-f013:**
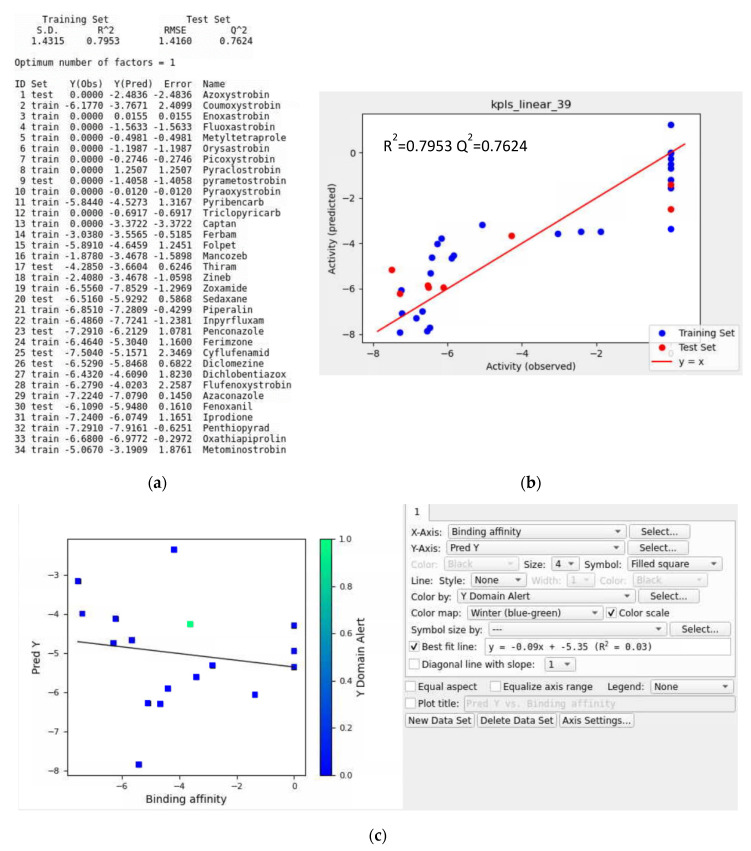
(**a**) Model report of the kpls_linear_39 model, (**b**) scatter plot providing the performance of the kpls_linear_39 model, and (**c**) scatter plot of the prediction set for the top QSAR models in Appendix A for *Plasmopara viticola* using a validation set.

**Table 1 microorganisms-11-01341-t001:** Recent in vitro studies on fungicide combination recommendations targeting *Plasmopara viticola* and *Botrytis cinerea*.

Study	Results	Limitations
Field efficacy of the combination of famoxadone and metalaxyl-M against *Plasmopara viticola* and the residue dynamics of the two fungicides in grapevine [17].	Formulation of 30% famoxadone with metalaxyl-M SC was effective against *Plasmopara viticola* [17].	Only a few fungicides were tested in the experiment.
Performance and phytotoxicity assessment of mancozeb 40% + azoxystrobin 7% OS against downy mildew of grapes in Maharashtra, India [18].	Formulation of 40% mancozeb with 7% azoxystrobin was effective against *Plasmopara viticola* [18].	Only a limited number of fungicides were tested over a period of two years.
Evaluation of synergistic activity and resistance development of the mixture of iprodione and fluopyram against *Botrytis Cinerea* [19].	Formulation of 80% iprodione with 20% fluopyram was effective against *Botrytis cinerea* [19].	Only two fungicides were tested in this research.
Bioefficacy of different fungicides against *Plasmopara viticola* and Erysiphe necator of grapes [20].	Formulations of 16.6% famoxadone with 22.1% cymoxanil, 10% famoxadone with 50% mancozeb, and 4.44% fluopicolide with 66.67% fosetyl-Al were recommended for *Plasmopara viticola* [20].	Experiments were limited to existing fungicide combinations on the market.
Synergy between Cu-NPs and fungicides against *Botrytis cinerea* [21].	Formulation of copper nanoparticles with fluazinam or thiophanate was effective against *Botrytis cinerea* [21].	The study was limited to a select few fungicides, and the long-term efficacy is unknown. The development of effective new nanoparticles would require many years, and the process of producing copper nanoparticles would be challenging.

**Table 2 microorganisms-11-01341-t002:** Resistance and mode of action information for fungicides selected in this study.

Fungicide	Resistance ^1^	Fungicide Type ^2^
Ubiqunol	NA	NA
Famoxadone	HR	QoI
Azoxystrobin	HR/R	QoI
Fenamidone	HR	QoI
Coumoxystrobin	HR	QoI
Flufenoxystrobin	HR	QoI
Enoxastrobin	HR	QoI
Pyraoxystrobin	HR	QoI
Picoxystrobin	HR	QoI
Metyltetraprole	HR	QoI
Fenaminstrobin	HR	QoI
Pyribencarb	HR	QoI
Dimoxystrobin	HR	QoI
Triclopyricarb	HR	QoI
Metominostrobin	HR	QoI
Pyrametostrobin	HR	QoI
Mandestrobin	HR	QoI
Fluoxastrobin	HR	QoI
Pyraclostrobin	HR	QoI
Orysastrobin	HR	QoI
Folpet	LR	PHT
Ferbam	LR	DTC
Captan	LR	PHT
Mancozeb	LR	DTC
Ametoctradin	HR/R	QoI
Thiram	LR	DTC
Zineb	LR	DTC

^1^ Resistance: NA, native; HR, high risk; LR, low risk for the resistance of fungicides. ^2^ Fungicide type: QoI, quinone outside inhibitor; DTC, dithiocarbamate; PHT, phthalimides.

## Data Availability

The data supporting the findings of this study are available from the correcponding author SF on request.

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
