# Peer review of "Identification of Fungicide Combinations Targeting Plasmopara viticola and Botrytis cinerea Fungicide Resistance Using Machine Learning"

_microorganisms, 2023, doi:10.3390/microorganisms11051341_

Round 1

Reviewer 1 Report

A MACHINE LEARNING COMBINED IN SILICO AP- 2 PROACH FOR THE IDENTIFICATION OF FUNGICIDE 3 COMBINATIONS TARGETING PLASMOPARA VITICOLA 4 AND BOTRYTIS CINEREA FUNGICIDE RESISTANCE

The paper is addressing a good field of research. But the paper needs some modifications. Below are the comments that needs to be taken into consideration.

1.      The title is too big make it more concise.

2.      The research problem is not explained in the abstract. Please write a line or 2 discussing the importance and significance of conducting this work

3.      Results are not shown in the abstract.

4.      The references starting in introduction is [9] it is not normal. It should start with ref [1] and continue in sequential order for ease of following the reference list. Please reshuffle the reference.

5.      What is the major significant contribution of this work. Please list them at the end of the introduction section.

6.      A propose Literature review section is needed for the readers to understand the state-of-the-art work done in this research problem. What is the research gap and how tis work is countering that gap? At the end of the literature review section please include a table summarizing the whole section with results, methodologies and shortcomings.

7.      A separate section should be dedicated for dataset.

8.      What is the methodology and technique applied in this research

9.      What are the performance metrics? How can it be justified that the results are good?

10.   Please include a conclusion section.

11.   Most of the reference cited are very old. Please cite references from 2023, and 2022 and discuss the work in details.

Reviewer 2 Report

In this study, a combination of molecular docking and QSAR algorithms was used to obtain information on the most potent inhibitory fungicide against wild-type fungal cytochrome b and its G143A mutation, respectively. The study has some merit, but the presentation of results must be improved.

References must be numbered in order of appearance in the text. Now the references start with no 9 followed by 19!

Materials and Methods section

1)      In this section, the authors should enumerate all the fungicides considered in this study and present a figure with their 2D formulas. In this manner, figures 14, 16, 22, 24, 30, 32, 38 and 40 are not necessary any more.

2)      Lines 108-109:From the following sentence „The 3D structures of ligands were obtained from ZINC15 or PubChem (all ligands are provided in supplementary materials) and generated in PDB format using online SMILES translator” I do understand what the authors retrieved from PubChem/Zinc , the SMILES formulas or the 3D structures? Because, if they retrieved the 3D structures, why they generated PDB files using SMILES translator?  

RESULTS section

1)      Figure 1B, in the legend it must be specified that ubiquinol is presented in red sticks and the cytochrome b in grey surface.

2)      Lines 177-179: I do not understsnd this sentence, I suppose that something is wrong. „Since the focus of this study was to identify fungicides that were effective against G143A mutation of the cytochrome b, a set of known antifungal agents were docked onto WT and G143A mutated types of Plasmopara viticola cytochrome b: the WT and G143A.”

3)      Table 1 corresponds better to Materials and Methods section.

4)      Please, explain better the figure 2. In its legend we should find information about the meaning of „level”, „mean” and letters from A to I. Similar for figures 3, 6, 7. In the legendes of these figures the word „select” must be replaced with „selected”.

5)      Figures 3, 6, 7: I do not understand what it is represented on the OY axe. I suppose that it is binding energy as it is specified further in Tables 3-13. If it is true, the word „score” must be replaced and  the measurements units for all the binding energies presented in figures and tables must be added. Moreover, the information regarding the binding energies obtained using molecular docking for the interactions of fungicides with the enzyme is presented in Figures 3, 6 and 7  and the same information is further presented in tables 3-13. It is redundant and the authors must find a way to present the information contained in Tables 3-13 such as to be syhthetic and do not repeat information that is already presented in figures. It will considerably reduce the number of tables and the typographical space.

6)      In the legend of Figure 4, please explain why THR residue is coloured in blue and the other residues are in yellow. I suppose that it corersponds to the polar character of THR, but it should be explained.

7)      Figure 5: the legend should explain the colour for every fungicide and that they are vizualized as sticks, whereas the cytocrome b is visualized as grey surface. The figure on the right is not at all explained. It should be mentioned what it is vizualized there.  Furthermore, in the legend text, the verb is missing.  

8)      Subsection 3.4 is more adequately to appear in the Materials and Methods section that in Results section.

9)      Figure 8 - the colours of the helices sorrounding the binding sites must be explained, or ar least what does it emphasize the right down images for both figures 8a and 8b. At least residues G143 and F129 should be emphasized in these pictures as the authors specify that “An analysis of the binding behavior of ametoctradin, pyraoxystrobin, mandestrobin, enoxastrobin and pyribencarb at the vicinity of the G143 and F129 residues of WT cytochrome b of Botrytis cinerea indicated that they all bound close to the two residues (Figure 8).”

10)   Figure 9: This pictures shows the interactions between ubiquinol and respectively pyraoxystrobin with WT cytochrome b, and with the mutated protein and not, so the authors should mention that there are interactions!

11)  Row 336, the word “spilt” should be replaced by “split”.

12)  Figures 10-13, 15, 17-21, 23, 25-29, 31, 33-37, 39 and 41 can be presented as supplementary materials. The manuscript is very long and hard to follow.

13)  I am not a native English speaker, but I find the manuscript hard to follow, there are many grammar errors and the language should be improved.

Round 2

Reviewer 1 Report

Comment number 5, 6 and 7 are not addressed.

Author Response

See attached response.

Response to Reviewers Round 2

Reviewer 1

  1. What is the major significant contribution of this work. Please list them at the end of the introduction section.

Response: We added a section to explain the above concern at the end of the Introduction section as follows:

The significance of this work is that, as of now, there is only limited information for researchers to follow on selecting fungicides for Plasmopara viticola and Botrytis cinerea once resistance to existing fungicides is suspected. Also, there is no objective way to suggest fungicide combinations to reduce the development of resistance or treat crops that have already developed fungicide resistance. This study aims to provide a thermodynamic-coupled machine learning strategy to identify and select antifungal agents from QoIs (high-risk group) combined with low-risk fungicides to form fungicide combination(s) that can mitigate fungicide resistance. This approach is based on docking selected fungicides from QoIs and low-risk fungicides with a homology model of cytochrome b to identify the fungicides with the highest affinity and evaluating the screened fungicides using QSAR models with machine learning statistical methods.

  1. A propose Literature review section is needed for the readers to understand the state-of-the-art work done in this research problem. What is the research gap and how this work is countering that gap? At the end of the literature review section please include a table summarizing the whole section with results, methodologies and shortcomings.

Response: We added a Literature Review section discussing the most recent studies, the current research gap, and the reasons behind in silico studies countering that gap. We also added a table summarizing the most recent work as follows:

Currently, only a few studies exist recommending fungicide combinations targeting Plasmopara viticola and Botrytis cinerea resistance as summarized in Table 1. There is a gap in knowledge on what fungicide combinations should be selected to treat crops that have already developed resistance and also what combinations would be the most logical to be used to prevent developing resistance. The approach we have taken in this study is to evaluate how fungicides interact with WT and mutated versions of the target protein(s) at a molecular level to screen highest binding fungicides that in turn will be recommended for field testing. 

Table 1. Recent in vitro studies on fungicide combination recommendations targeting Plasmopara viticola and Botrytis cinerea.

Study

Results

Limitations

Field efficacy of the combination of famoxadone and metalaxyl-M against Plasmopara viticola and the residue dynamics of the two fungicides in grapevine [17]

Formulation of 30% famoxadone with metalaxyl-M SC was effective against Plasmopara viticola [17].

Only a few fungicides were tested in the experiment.

Performance and phytotoxicity assessment of mancozeb 40% + azoxystrobin 7% OS against downy mildew of grapes in Maharashtra, India [18]

Formulation of 40% mancozeb with 7% azoxystrobin was effective against Plasmopara viticola [18].

Only a limited number of fungicides were tested over a period of two years.

Evaluation of synergistic activity and resistance development of the mixture of iprodione and fluopyram against Botrytis Cinerea [19]

Formulation of 80% iprodione with 20% fluopyram was effective against Botrytis cinerea [19].

Only two fungicides were tested in this research.

Bioefficacy of different fungicides against Plasmopara viticola and Erysiphe necator of grape [20]

Formulations of 16.6% famoxadone with 22.1% cymoxanil, 10% famoxadone with 50% mancozeb, and 4.44% fluopicolide with 66.67% fosetyl-Al were recommended for Plasmopara viticola [20].

Experiments were limited to existing fungicide combinations in the market.

Synergy between Cu-NPs and fungicides against Botrytis cinerea [21]

Formulation of copper nanoparticles with fluazinam or thiophanate was effective against Botrytis cinerea [21].

The study was limited to a select few fungicides and the long-term efficacy is unknown. The development of effective new nanoparticles required many years and the process of producing copper nanoparticles would be challenging.

Select studies on fungicide combinations are shown in Table 1. A key limitation is that, since the studies were experimentally based, only a limited number of fungicides could be tested. Moreover, the long-term efficacy of these combinations is yet to be determined. A key advantage of in silico-based methods is their ability to screen a large number of fungicides simultaneously based on rational molecular-level information and select those with the highest promise for field testing. According to the existing literature, no studies have provided guidance for the selection of fungicide combinations based on molecular structures and their affinity to the target active site.

  1. A separate section should be dedicated for dataset.

Response: We have created a separate section (4. DATASET) at line 595-614 for dataset. We moved all the data pertinent to QSAR work to this section.

We also checked for spelling errors throughout the manuscript while revising the conclusions further to match the results.

Reviewer 2 Report

The revised version of this manuscript is considerably improved. However, there are still few points that need to be better explained. 

Author Response

See attached response.
